# An End-to-End Real-World Camera Imaging Pipeline

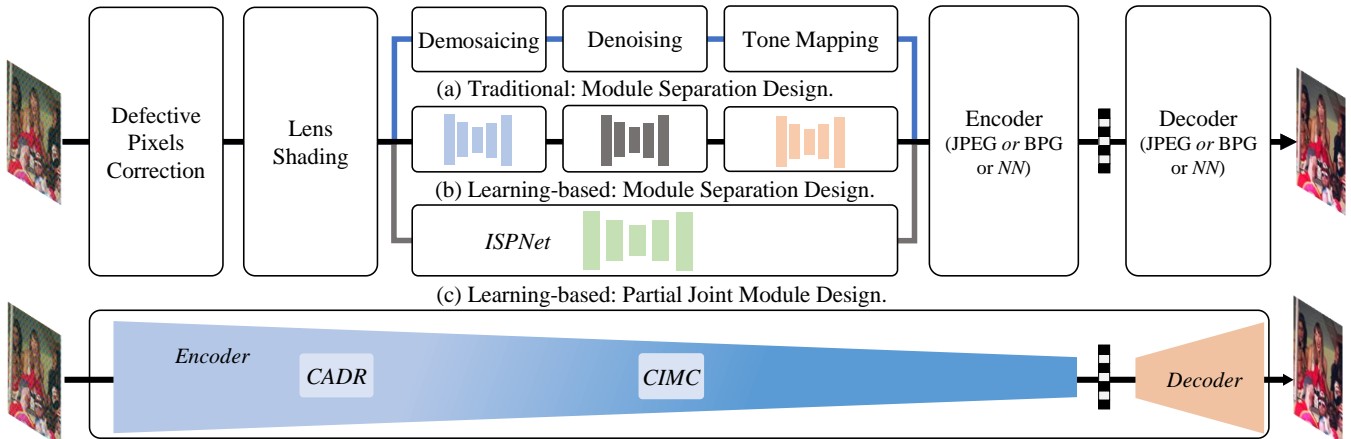

**Figure 1: (a)Traditional Design: Implements each step of Image Signal Processing (ISP) separately using conventional algorithms. (b) Learning-based Separated Design: Employs neural networks to individually address each phase of the ISP process. (c) Learning-based Partial Joint Design: Develop an ISPNet to unify operations such as demosaicing, and tone mapping. (d) RealCamNet: We propose an end-to-end camera imaging framework that categorizes ISP operations into coordinate-independent and coordinate-dependent groups. CIMC and CADR are designed to perform tasks like demosaicing and image feature compression and to restore coordinate-dependent image distortions (e.g., vignetting, dark shadows), respectively.**

## ABSTRACT

Recent advances in neural camera imaging pipelines have demonstrated notable progress. Nevertheless, the real-world imaging pipeline still faces challenges including the lack of joint optimization in system components, computational redundancies, and optical distortions such as lens shading. In light of this, we propose an end-to-end camera imaging pipeline (RealCamNet) to enhance real-world camera imaging performance. Our methodology diverges from conventional, fragmented multi-stage image signal processing towards end-to-end architecture. This architecture facilitates joint optimization across the full pipeline and the restoration of coordinate-biased distortions. RealCamNet is designed for high-quality conversion from RAW to RGB and compact image compression. Specifically, we deeply analyze coordinate-dependent optical distortions, e.g., vignetting and dark shading, and design a novel Coordinate-Aware Distortion Restoration (CADR) module to restore coordinate-biased distortions. Furthermore, we propose a Coordinate-Independent Mapping Compression (CIMC) module to implement tone mapping and redundant information compression.

**Unpublished working draft. Not for distribution.**

Existing datasets suffer from misalignment and overly idealized conditions, making them inadequate for training real-world imaging pipelines. Therefore, we collected a real-world imaging dataset. Experiment results show that RealCamNet achieves the best rate-distortion performance with lower inference latency.

## CCS CONCEPTS

• **Computing methodologies** → **Computational photography**; **Computational photography**; **Image compression**.

## KEYWORDS

Camera imaging, deep neural network, image compression, image signal processing

## 1 INTRODUCTION

Efficient imaging and compression technologies are paramount to the internet and multimedia industries. With the exponential increase in digital content, techniques that reduce file sizes while maintaining image quality have become crucial. These advancements not only enhance the user experience by ensuring fast image and video loading but also help save storage space and reduce data transmission costs. Moreover, such technologies are indispensable for supporting advanced applications like autonomous driving and remote sensing, driving technological innovation, and meeting the growing demands for multimedia processing. Therefore, developing more efficient imaging and compression methods is vital for propelling industry progress and meeting the needs of modern technology.

In traditional real-world camera imaging pipelines, complex and proprietary hardware processes are employed for image signal processing (ISP), encompassing steps such as denoising, demosaicing, tone mapping, and image compression, as shown in Fig.1 (a). The pipeline's step-by-step design, where each process is designed separately, leads to error accumulation and prevents the system from achieving an optimal state.

Most traditional methods derive solutions at each step of the ISP pipeline using heuristic approaches[14, 28, 39], thus requiring the adjustment of numerous parameters. In addition, errors will be accumulated in the ISP processing flow, affecting imaging quality.

Deep neural networks have developed rapidly in the past decade, and are used in image classification[33, 35], object detection[26, 44], image and video enhancement[2, 25, 38, 43, 45], natural language processing[9, 23, 36] and other fields have played an important role. Researchers have also proposed a series of image signal processing methods based on neural networks. In recent years, researchers[15, 22] have attempted to construct camera imaging systems based on deep neural networks, designing models for denoising, demosaicing, tone mapping, and compression, significantly enhancing overall system performance. However, constructing separate neural networks for each function (e.g., denoising, demosaicing) prevents the system from being jointly optimized and introduces computational redundancy, as illustrated in Fig.1 (b).

Therefore, some methods[18, 19, 47] have emerged to try to build a single neural network to complete the functions required by ISP. This is a good idea, but because the situation considered is too simple, the designed ISPNet cannot be realistically applied.

To address these issues, we propose the RealCamNet framework, which integrates the real camera image signal processing and image compression processes into an end-to-end deep neural network framework. This unified approach not only reduces computational redundancy but also enhances both image quality and compression efficiency through the joint optimization of network parameters, a feat unattainable by previously isolated design methods.

Our starting point was to construct a neural network-based real-world camera pipeline that simulates common functions found in real-world applications, building a reliable and practical end-to-end imaging system. To this end, we analyzed the principles and flaws in each step of the imaging and compression process.

Our RealCamNet's overall framework is shown in Fig.1 (d), where we designed an end-to-end deep neural network to implement the RAW->Bitstream->RGB imaging compression process. The past cascaded framework depicted in Fig.1 (a) and Fig.1 (b) leads to problems such as error accumulation from each module and potentially suboptimal global results. Fig.1 (c) constructing a neural network directly implements operations such as demosaicing and tone mapping, but the overall joint optimization is still not achieved.

In our RealCamNet, we designed global and local perception imaging pipeline modules to simulate processes in the imaging pipeline, such as global and local tone mapping, denoising, demosaicing, and image feature compression.

Our investigation into camera imaging technologies has identified digital signal distortions stemming from optical system imperfections. In optical camera imaging, Coordinate-dependent distortions significantly impact image quality due to the inherent characteristics of the optical system and sensor. These include

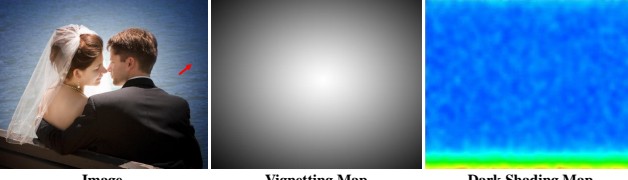

**Image**  **Vignetting Map**  **Dark Shading Map**

**Figure 2: Coordinate-dependent distortion. The left side is the captured image, and the right side is the vignetting distribution map. The middle of the picture on the left is brighter.**

Coordinate-dependent noise resulting from sensor sensitivity variations or uneven optical system illumination, and vignetting, a reduction in edge brightness caused by lens design and light entry angles. As well as dark shading caused by problems such as uneven heating of CMOS due to the superposition of components. The vignetting and dark shading is shown in Fig.2. To mitigate these distortions and improve image quality, sophisticated image processing is required. We introduce a streamlined, effective coordinate-aware distortion correction module designed to identify and rectify various Coordinate-biased distortions, enhancing visual perception and imaging quality.

Previous RAW-RGB datasets either have too simple ISP processes (only include a few processes such as demosaicing), or have misalignment issues and cannot be used to train camera imaging pipelines that can be used in real scenes. To train our RealCamNet, we construct a new RAW-RGB dataset to directly learn the complex ISP process of real cameras.

In summary, our contributions are fourfold:

- We constructed a deep neural network imaging system and demonstrated the performance improvements from end-to-end joint optimization.
- We deeply analyze coordinate-dependent optical distortions, e.g., vignetting and dark shading, and design a novel Coordinate-Aware Distortion Restoration (**CADR**) module to repair coordinate-biased distortions.
- We designed a Coordinate-Independent Mapping Compression (**CIMC**) module to implement global and local tone mapping, denoising, demosaicing, and image feature compression functions, thereby reducing computational redundancy.
- We built a new real-world imaging dataset, providing a benchmark for the unified end-to-end neural imaging pipeline.

## 2 PRELIMINARY

### 2.1 Traditional Image Signal Processing

Traditionally, the Image Signal Processing (ISP) is responsible for reconstructing RGB images from RAW captures. In conventional camera pipelines, complex and proprietary hardware processes are employed for image signal processing. This process encompasses several steps, including denoising, demosaicing, defect pixel correction, and tone mapping[6, 11, 13, 27, 28, 40], among others. Each module requires individual tuning and optimization, with consideration for the adjustment of cascading errors.

**Figure 3: Ours Framework. The encoder $E$ of RealCamNet proposes $CADR$ to restorate coordinate-related distortion and builds $CIMC$ to complete coordinate-independent functions (such as global and local tone mapping, denoising, and feature compression). The decoder $D$ of RealCamNet proposes $CSA$ to decode the decoded features and restore the RGB image. $LFT$ is local feature transform, and $GFT$ is global feature transform.**

## 2.2 Learning-based Image Signal Processing

Advancements in deep learning have led researchers to enhance ISP pipelines via neural networks. Localized network solutions segment functions such as tone mapping[34] and denoising[16, 22], facilitating module decoupling yet complicating the ISP's holistic optimization. Comprehensive neural network approaches replace the entire ISP pipeline, streamlining computations and enabling system-wide optimization[10, 20, 29, 48].

## 2.3 Learning-based Image Compression

In learning-based image compression, Ballé et al.[3] pioneered the use of an encoder-decoder architecture with entropy coding for latent feature representation, enabling model optimization. Progress by Ballé et al.[4] reduced latent feature redundancy through adaptive means and variances. Cheng et al.[7] improved accuracy using Gaussian Mixture Models (GMM) with hyper-priors for GMM parameters. Minnen et al.[5, 12, 31, 32] furthered this with auto-regressive models in entropy coding, decreasing redundancy. However, the sequential nature of auto-regressive methods limited parallel processing, prompting developments like grouped Hyper channel-wise[30] and checkerboard[17] auto-regressive models to enhance speed without losing compression efficiency.

## 3 METHODOLOGY

### 3.1 Problem Formulation

The Camera Imaging Pipeline converts a RAW image into a compressed RGB format through a structured process. This process involves three primary stages: conversion from RAW to RGB, compression of the RGB image into a bitstream, and decompression of the bitstream to an RGB image. The aim is to optimize storage and transmission efficiency while preserving image quality. Formally,

the pipeline is described by the following sequence of operations:

$$
\begin{aligned}
r &= \mathcal{R}(x) \\
b &= C(r) \\
o &= D(b)
\end{aligned}
\tag{1}
$$

In this formulation, $x$ is the input RAW image, $\mathcal{R}$ denotes the RAW-to-RGB conversion, $r$ represents RGB image, $C$ represents the compression into a bitstream, $D$ signifies the decompression to RGB, $b$ is bitstream, and $o$ is the final RGB output. The end-to-end pipeline encapsulates the entirety of the imaging process, from initial capture to final output, in a single integrated model.

### 3.2 Architecture of RealCamNet

To enhance the performance of camera imaging pipelines, we propose *RealCamNet*, a novel end-to-end pipeline designed for real-world camera imaging. This pipeline integrates both encoder and decoder components to facilitate effective image processing. The operational framework of *RealCamNet* is formalized as follows:

$$
\begin{aligned}
y &= \mathcal{E}(x; \phi) \\
\hat{y} &= Q(y) \\
o &= \mathcal{D}(\hat{y}; \theta)
\end{aligned}
\tag{2}
$$

where $x$ denotes the input RAW image, and $o$ represents the output reconstructed RGB image, $\phi$ represents the weights of the encoder $\mathcal{E}$, while $\theta$ denotes the weights of the decoder $\mathcal{D}$.

The encoder $\mathcal{E}$ in our pipeline is tasked with performing RAW to RGB conversion and RGB compression encoding. It comprises two principal modules:

(1) The Coordinate-Aware Distortion Restoration (**CADR**) module, is aimed at correcting coordinate-dependent distortions such as vignetting, noise, and dark shading, which result

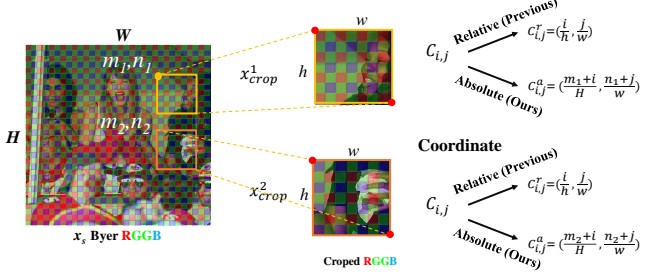

Figure 4: Compared with previous methods that can only encode the relative coordinates of the cropped image, our method calculates the absolute coordinates of the cropped RAW image. Therefore our method can recover fixed position-type distortion in the image.

from optical and manufacturing defects. The term 'Coordinate' refers to the pixel coordinates in the original RAW image.

(2) The Coordinate-Independent Mapping Compression (**CIMC**) module, is responsible for executing spatially invariant operations including global and local tone mapping, denoising, demosaicing, and image feature compression.

For the encoding process, the RAW image $x_{raw}$ undergoes UnpixelShuffle to achieve channel stacking, converting the spatial dimension RGGB to the channel dimension RGGB image $x_s$. Subsequently, $x_s$ and the coordinate code $x_c$ are fed into the **CADR** module to perform position-related restoration, thereby addressing distortions like vignetting and dark shading. A Color Prior Extraction (**CPE**) module is then utilized to extract color prior information, aiding the **CIMC** module in performing the tone mapping and compression processes from RAW to RGB. A series of **CIMC** modules are employed for image feature compression and tone mapping, followed by the encoding of tone-mapped and compressed latent features into a bitstream via the Entropy Model.

In the decoding phase, the Entropy Model first converts the bitstream back into latent features. These features are processed through the concatenated Channel Spatial Attention (**CSA**) module, which recovers detailed image information from the compact latent features. Finally, the UnpixelShuffle module is used to reconstruct the RGB image from these features.

This full framework is shown in Fig. 3, illustrates the detail of *RealCamNet*, which is structured into three main components: encoder, entropy model, and decoder, all of which are jointly trained in an end-to-end manner.

## 3.3 Coordinate-Aware Distortion Restoration

We present the detailed architecture of the proposed Coordinate-Aware Distortion Restoration (CADR) module, depicted in Fig. 3 (b). The CADR module achieves effective coordinate awareness by integrating the current pixel coordinates, thus facilitating the learning of coordinate-dependent distortion restoration.

Initially, we introduce the previous coordinate calculate method[48]. Since it is impractical to train the network using full-size RAW $x_s$ $in R^{H,W,4}$ images, typically exceeding $4000 \times 6000$ dimensions.

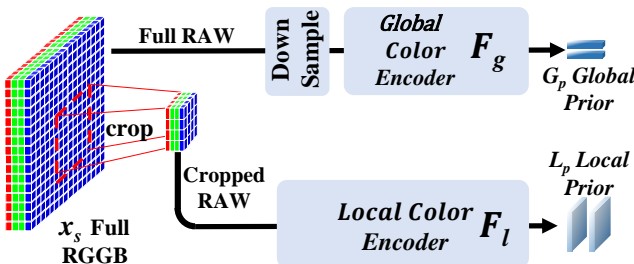

Figure 5: The detail of the Color Prior Encoder (CPE).

Previous methods pre-crop the RAWs into a small cropped RAW dataset, represented as $x_{crop} \in \mathbb{R}^{h,w,4}$, which is suitable for training neural networks. For the cropped image $x_{crop} \in \mathbb{R}^{h,w,4}$, this method ascertains the pixel relative coordinates for each point and stores this coordinate map $c^r \in \mathbb{R}^{h,w,2}$. The calculation method is shown in Eq.3:

$$c^r_{i,j} = (\frac{i}{h}, \frac{j}{w}) \tag{3}$$

As illustrated in Fig.4, $c^r_{i,j}$ details are presented. Consider that $x^1_{crop}$ and $x^2_{crop}$ are randomly cropped from the same RAW image. For left-top coordinate $(m_1, n_1)$ in $x^1_{crop}$ and left-top coordinate $(m_2, n_2)$ in $x^2_{crop}$, the relative coordinates are consistently $c^{0,0}_r = (0, 0)$. However, the absolute coordinates are distinct, with $c^{0,0}_a = (\frac{m_1}{H}, \frac{n_1}{W})$ for the first and $c^{0,0}_a = (\frac{m_2}{H}, \frac{n_2}{W})$ for the second. Clearly, $c_r$ fails to capture the true coordinates, whereas the $c_a$ successfully retains the actual positional information. This straightforward design facilitates effective distortion restoration.

To address this limitation, we propose a simple method of coordinate embedding to capture the genuine global coordinates of the RAW image. This entails modifying the coordinate computation approach. We first ascertain the coordinates of the top-left pixel of $x_{crop}$ relative to $x_s$, denoted as $< m, n >$. Next, calculate the absolute coordinates $c^a_{i,j}, c^a_{i,j}$ records the coordinate position of each pixel relative to the original RAW, $c^a_{i,j} See Eq.4 for calculation details of$. This method of coordinate embedding allows our CADR to accurately perceive the pixel's spatial location, thereby effectively restoring coordinate-related distortions.

$$c^a_{i,j} = (\frac{i+m}{H}, \frac{j+n}{W}) \tag{4}$$

Upon acquiring the absolute coordinate $c^a_{i,j}$, we integrate the coordinate data into the encoder to achieve coordinate-aware distortion restoration. Initially, the stacked RGGB image $x_{crop}$ is input into a $3 \times 3$ convolution layer to extract latent feature $x_h$. Concurrently, the coordinate information $u^n_{i,j}, v^n_{i,j}$ is processed through a Convolution-ReLU sequence to derive the potential coordinate embeddings $x_e$. To facilitate coordinate-aware enhancement, the potential coordinate embedding $x_e$ is multiplied by $x_s$ to obtain enhanced features $x_o$. The entire process is encapsulated in Eq.5.

$$\begin{aligned} x_h &= \text{Conv}(x_s) \\ x_e &= \text{ReLU}(\text{Conv}(c^a)) \\ x_o &= x_h \cdot x_e \end{aligned} \tag{5}$$

## 3.4 Color Prior Encoder

Figure 5 show the two-fold structure of our Color Prior Extraction (CPE) module: Global prior extraction and Local prior extraction. The global prior extraction process involves downsampling the original $4000 \times 4000$ RAW image to a $256 \times 256$ representation, preserving global features while reducing computational load, as defined by:

$$G_p = \mathcal{F}_g(\text{downsample}(x_s)), \tag{6}$$

where $x_s$ is the original RAW image, $\mathcal{F}_g$ represents the global color encoder, and $G_p$ denotes the global color prior.

In parallel, local prior extraction focuses on the details within a specific cropped region by applying the local color encoder to the cropped $256 \times 256$ RGGB image, described by:

$$L_p = \mathcal{F}_l(x_{crop}), \tag{7}$$

where $x_{crop}$ symbolizes the cropped RAW image, $\mathcal{F}_l$ is the local color encoder, and $L_p$ represents the local color prior. The details of $\mathcal{F}_l$ and $\mathcal{F}_g$ are introduced in the appendix.

## 3.5 Channel-Spatial Attention

The Channel-Spatial Attention (CSA) mechanism forms an integral part of the Coordinate-independent Mapping Compression (CIMC) module. Inspired by the principles of intra-frame coding in traditional video compression, the CSA is designed to identify and leverage non-local redundancy within feature representations, promoting more efficient compression. Within the encoder, CSA utilizes an attention mechanism to aggregate information across both spatial and channel dimensions, allowing for the reduction of superfluous data and the distillation of features into compact representations that are conducive to improved compression efficacy. During the decoding phase, CSA plays a crucial role in reconstructing non-local reference features from the compressed feature set. This functionality is vital for restoring the quality of the reconstructed image and ensuring the fidelity of the output relative to the original input.

*3.5.1 Detail of CSA.* We delve into the computational intricacies of the CSA (Channel Spatial Attention) as follows. The initial input feature $x_{in}$ undergoes a transformation via a convolution layer:

$$x' = \text{Conv}(x_{in}) \tag{8}$$

Subsequently, the processed feature $x'$ is bifurcated into two distinct components $x_1$ and $x_2$:

$$x_1, x_2 = \text{Split}(x') \tag{9}$$

The first component, $x_1$, is subjected to the Channel-Wise Residual Attention (CWRA) yielding $x_3$:

$$x_{ca} = \text{CWRA}(x_1) \tag{10}$$

Simultaneously, the second component, $x_2$, traverses through the Spatial-Wise Attention (SWA) module, producing $x_{sa}$:

$$x_{sa} = \text{SWA}(x_2) \tag{11}$$

In our spatial-wise attention module, we employ the Swin Transformer. The fusion of $x_{sa}$ and $x_{ca}$ is accomplished via concatenation, followed by a convolutional layer to get output feature $x_o$:

$$x_o = \text{Conv}(\text{Concat}(x_{sa}, x_{ca})) \tag{12}$$

This approach synergizes channel and spatial attention, enhancing feature refinement and facilitating the transformation of RAW image features into a compact RGB latent representation. The result is an improved reconstruction quality with reduced redundancy, essential for efficient image processing tasks.

## 3.6 Coordinate-Independent Mapping Compression

*3.6.1 Overview of CIMC.* The Coordinate-independent Mapping Compression (CIMC) module is designed to compress RAW image features into compact RGB latent representations. Employing attention mechanisms and feature transformations, CIMC aims to enhance reconstruction quality and reduce data redundancy.

CIMC consists of three key components: the Channel-Spatial Attention (CSA) module, the Local Feature Transformation (LFT) module, and the Global Feature Transformation (GFT) module. LFT and GFT can use the color prior extracted by the CPE module for tone mapping. CSA promotes effective compression of latent features, as shown in Fig.3 (d).

*3.6.2 Details of CIMC.* Incorporating both GFT and LFT within the Encoder, CIMC extends CSA to perform global and local tone mapping essential for the camera imaging pipeline. Positioned after the CSA channel attention stage, LFT serves to refine features (Fig. 3):

$$x_{ca} = LFT(x_{ca}, L_p) \tag{13}$$

Subsequent iterations of CSA and LFT result in the feature $x_c$:

$$x_c = LFT(CSA(LFT(CSA(x_h), L_p)), L_p) \tag{14}$$

The concluding phase entails the transformation of $x_c$ by GFT to yield the output $x_o$, representing the compressed RGB latent features:

$$x_o = GFT(x_c, G_p) \tag{15}$$

where $G_p, L_p$ is the output of the CPE module. The calculation of GFT and LFT are both $y = \alpha x + \beta$. The difference is that $\alpha$ and $\beta$ are globally or locally adaptive. This module allows CIMC to enhance color fidelity through integrated local and global feature transformations.

## 4 EXPERIMENT

### 4.1 Implementation Details

To train RealCamNet, we created a comprehensive RAW-RGB dataset. Our dataset, with 4507 RAW-RGB image pairs at $6000 \times 4000$ resolution, covers diverse scenes including animals, landscapes, and architecture. For evaluation, we employ 450 pairs, with the remainder for training. Further dataset specifics are in the Appendix. For optimization, we employed the Adam optimizer [24, 42] with an initial learning rate of $1 \times 10^{-4}$, implementing a multi-step rate reduction strategy. The RD-formula loss, as per [7], was used, testing $\lambda$ values within $[0.1, 0.025, 0.01, 0.005]$ to accommodate different bitrates. The training was conducted on an i9-13900K CPU and an RTX 4090 GPU, with batch size set at 8.

### 4.2 Metrics

Evaluation metrics used in our study include PSNR (Peak Signal-to-Noise Ratio) [37], MS-SSIM (Multi-Scale Structural Similarity Index)

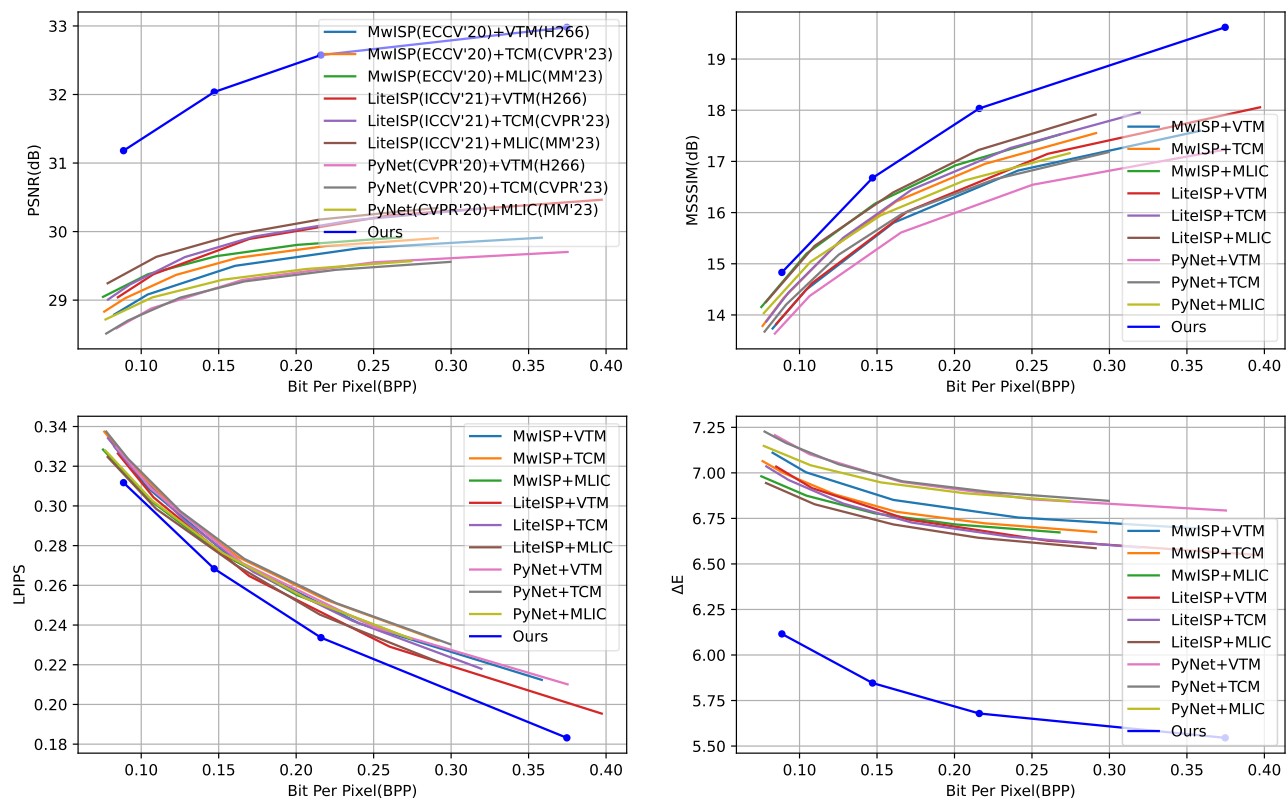

**Figure 6: Quantitative Rate-Distortion curve results.**

**Table 1: Quantitative results. We compare with state-of-the-art ISPNet and image compression methods, including learning-based methods: PyNet(CVPR'20)[19], LiteISPNet(ICCV'21)[47], MwISPNet(ECCV'20)[18], MLIC(ACMMM'23)[21], TCM(CVPR'23)[30] and the most advanced traditional image compression method VTM/H.266. We show BD-Rate, BD-PSNR, BD-MSSSIM, BD-$\Delta E$ and BD-LPPHS for all methods. We use PyNet+VTM as anchor.**

| Method | Params (M)↓ | FLOPs (G)↓ | Enc Time (s)↓ | Dec Time (s)↓ | BD - Rate↓ | BD- PSNR(db)↑ | BD- MSSSIM(db)↑ | BD- LPIPS↓ | BD-$\Delta E$↓ |
|---|---|---|---|---|---|---|---|---|---|
| PyNet+VTM[1, 19] | - | - | 196.94 | 0.1433 | 0.0000 | 0.0000 | 0.0000 | 0.0000 | 0.0000 |
| PyNet+TCM[19, 30] | 92.51 | 2354 | 0.1582 | 0.1405 | -12.1248 | -0.0286 | 0.3270 | 0.0031 | 0.0079 |
| PyNet+MLIC[19, 21] | 164.03 | 3108 | 0.1695 | 0.2155 | -20.2375 | 0.0776 | 0.5603 | -0.0025 | -0.0314 |
| MwISP+VTM[1, 18] | - | - | 196.93 | 0.1433 | -10.2986 | 0.2262 | 0.2868 | -0.0012 | -0.1071 |
| MwISP+TCM[18, 30] | 74.18 | 1026 | 0.1615 | 0.1405 | -20.7415 | 0.3292 | 0.6163 | 0.0022 | -0.1658 |
| MwISP+MLIC[18, 21] | 145.7 | 1780 | 0.1727 | 0.2155 | -28.7059 | 0.4408 | 0.8593 | -0.0032 | -0.2076 |
| LiteISP+VTM[1, 47] | - | - | 196.93 | 0.1433 | -12.5324 | 0.5880 | 0.3900 | -0.0052 | -0.2022 |
| LiteISP+TCM[30, 47] | 53.98 | 940 | 0.1522 | 0.1405 | -22.0966 | 0.5911 | 0.7029 | -0.0010 | -0.2094 |
| LiteISP+MLIC[21, 47] | 125.5 | 1694 | 0.1635 | 0.2155 | -29.5519 | 0.7079 | 0.9223 | -0.0057 | -0.2513 |
| **Ours** | **49.01** | **357** | **0.0703** | **0.0592** | **-39.0842** | **2.9603** | **1.6392** | **-0.0162** | **-1.1709** |

[41], $\Delta E$ (a measure of color difference) [8], and LPIPS (Learned Perceptual Image Patch Similarity) [46]. PSNR measures image reconstruction quality, MS-SSIM evaluates image fidelity across multiple scales, $\Delta E$ assesses color accuracy, and LPIPS quantifies perceptual similarity using deep learning models. For clarity, MS-SSIM was converted to $-10 \log_{10}(1 - \text{MS-SSIM})$. The performance

across metrics was gauged using Bjøntegaard Delta (BD) metrics, encompassing BD-PSNR, BD-MSSSIM, BD-$\Delta E$, and BD-LPIPS.

## 4.3 Rate-Distortion Performance

To assess the efficacy of our newly proposed RealCamNet, we juxtapose its performance with the leading multi-stage separation schemes. The comparative analysis encompasses two principal

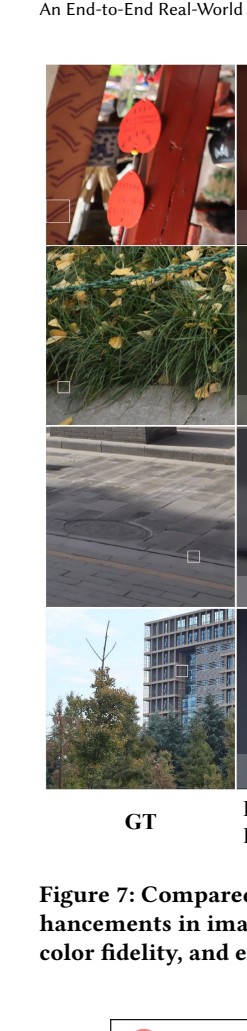
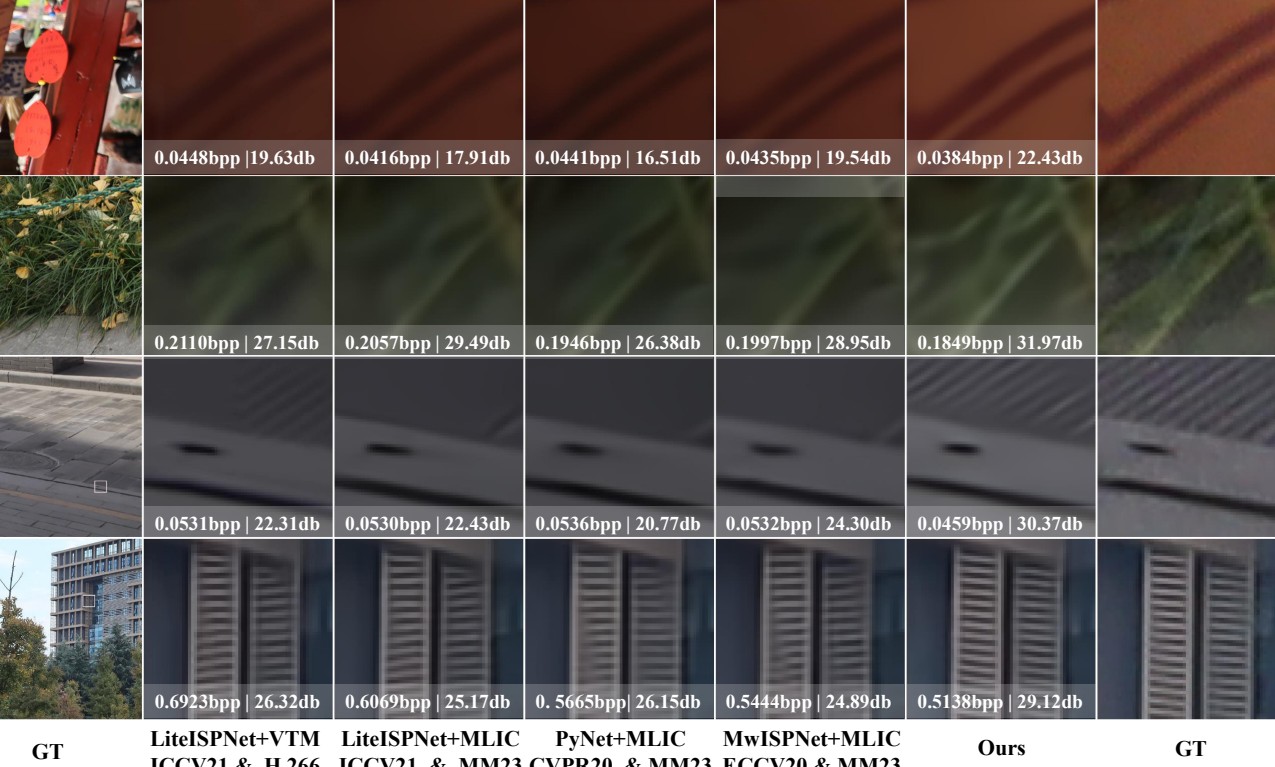

| | 0.0448bpp \|19.63db | 0.0416bpp \| 17.91db | 0.0441bpp \| 16.51db | 0.0435bpp \| 19.54db | 0.0384bpp \| 22.43db | |
| | 0.2110bpp \| 27.15db | 0.2057bpp \| 29.49db | 0.1946bpp \| 26.38db | 0.1997bpp \| 28.95db | 0.1849bpp \| 31.97db | |
| | 0.0531bpp \| 22.31db | 0.0530bpp \| 22.43db | 0.0536bpp \| 20.77db | 0.0532bpp \| 24.30db | 0.0459bpp \| 30.37db | |
| | 0.6923bpp \| 26.32db | 0.6069bpp \| 25.17db | 0.5665bpp\| 26.15db | 0.5444bpp \| 24.89db | 0.5138bpp \| 29.12db | |
| **GT** | **LiteISPNet+VTM** ICCV21 & H.266 | **LiteISPNet+MLIC** ICCV21 & MM23 | **PyNet+MLIC** CVPR20 & MM23 | **MwISPNet+MLIC** ECCV20 & MM23 | **Ours** | **GT** |

**Figure 7: Compared with the state-of-the-art methods, our framework, optimized end-to-end, demonstrates significant enhancements in imaging systems' performance. It offers comprehensive benefits, encompassing reduced bit rate, augmented color fidelity, and elevated PSNR.**

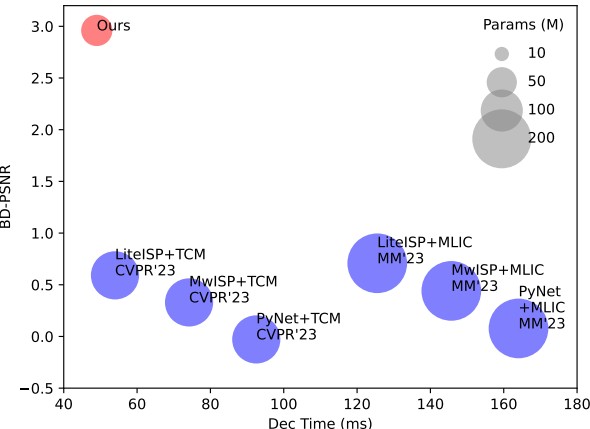

**Figure 8: Results of performance and inference time.**

pipelines: (1) a learning-based ISPNet combined with the Universal Video Coding (VVC) framework, and (2) ISPNet integrated with a learning-based image compression approach. Fig.6 delineates the rate-distortion outcomes across our test dataset. A thorough evaluation is conducted employing a suite of metrics—peak signal-to-noise ratio (PSNR), multi-scale structural similarity index (MS-SSIM),

color fidelity ($\Delta E$), and learning-based perceptual image patch similarity (LPIPS)—to gauge both the objective and perceptual quality, thus offering a holistic assessment of the varied pipelines.

With PyNet+VTM set as the anchor, we delve into the performance nuances of each pipeline using diverse metrics, calculating the Bjøntegaard Delta (BD) metrics across the metrics-BPP curve, inclusive of BD-PSNR, BD-MSSSIM, BD-$\Delta E$, and BD-LPIPS. The empirical data reveal that our RealCamNet, at an equivalent bitrate, transcends the conventional SOTA multi-stage, learning-based pipelines, marking improvements of 2.26dB in PSNR, 0.71dB in MS-SSIM, 0.01 in LPIPS, and 0.9187 in $\Delta E$. Additionally, when pitted against the learning-based ISPNet in conjunction with the Cascaded Universal Video Coding (VVC) framework, our method consistently exhibits superior performance in metrics like PSNR at comparable bitrates, underscoring the robustness and stability of our proposed approach.

To elucidate the performance of RealCamNet, we computed the BD-Rate based on the rate-distortion curve, revealing a significant enhancement over the best-existing pipeline, with a BD-Rate improvement of 9.53%. These findings, presented in Table 1, underscore the comprehensive performance superiority of our approach on the test set.

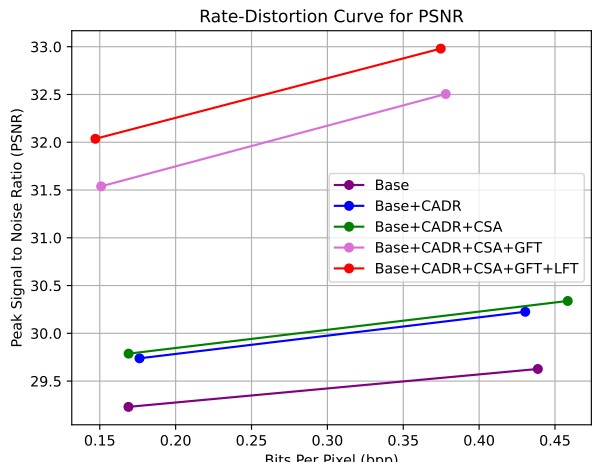

Figure 9: Ablation in Rate-Distortion curve.

## 4.4 Ablation Studies of CIMC

In Fig.9, we assess the efficacy of the **CIMC** module, which comprises CSA, LFT, and GFT components. The **CIMC** module enhances tone mapping through novel LFT and GFT while optimizing feature compression with CSA. Quantitative analysis of the **CIMC** and its individual CSA, LFT, and GFT components is presented in Table 2.

## 4.5 Ablation Studies of CADR

To demonstrate the effectiveness of our proposed Coordinate-Aware Distortion Restoration module **CADR**, we compared the model without the **CADR** module with the model adding the **CADR** module. The results are shown in Fig.9. The addition of our **CADR** module brought huge improvements. We show the quantitative gain on BD-PSNR in Table 2.

## 4.6 Complexity and Qualitative Results.

We compare the computational complexity of different methods. As shown in Table6 and Figure8, our method can decode images with a resolution of $1024 \times 1024$ at an inference speed of 16.8fps. Our approach not only achieves superior performance but also demonstrates lower computational complexity and faster inference speed. This dual advantage underscores the method's efficiency, marrying high-quality results with expeditious processing.

Table 2: Quantitative ablation results. Results are presented with 'Base' serving as the benchmark anchor.

| Base | CADR | CSA | GFT | LFT | Params (M) | FLOPs (G) | BD-PSNR(db) |
|------|------|-----|-----|-----|------------|-----------|-------------|
| ✓ | | | | | 48.511 | 79.810 | 0.0000 |
| ✓ | ✓ | | | | 48.561 | 83.048 | 0.4946 |
| ✓ | ✓ | ✓ | | | 46.071 | 71.059 | 0.7475 |
| ✓ | ✓ | ✓ | ✓ | | 47.290 | 77.438 | 2.6545 |
| ✓ | ✓ | ✓ | ✓ | ✓ | 49.010 | 89.176 | 3.1716 |

## 4.7 Visual Results

Fig. 7 presents a visual comparison of the reconstructed images using our method against Pipeline-1 (ISPNet coupled with the classic VVC standard, VTM 12.1) and Pipeline-2 (ISPNet integrated with a learning-based compression network). Our method exhibits superior performance in preserving intricate textures, as evidenced by clearer feather outlines, and achieves enhanced color fidelity.

## 4.8 Receptive Field Analysis

We analyze the effectiveness of each module by calculating the effective receptive field. The analysis of receptive fields for different inputs indicates that the model's output for Global downsamples RAW encompasses a global receptive field(Fig.10 (b)), thus facilitating the extraction of a global color before the entire image. Furthermore, the output for Cropped RAW exhibits a non-local receptive field(Fig.10 (c)), demonstrating that CSA (Channel Spatial Attention) is capable of harnessing non-local information to enhance the compactness of latent. In contrast, coordinate embedding is local to the input receptive field(Fig.10 (d)) because perceiving the coordinate information of the current position enables high-quality position-dependent distortion restoration.

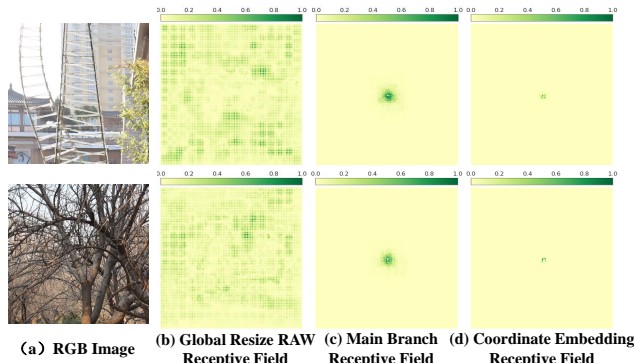

(a) RGB Image    (b) Global Resize RAW Receptive Field    (c) Main Branch Receptive Field    (d) Coordinate Embedding Receptive Field

Figure 10: Receptive field analysis. Our global branch can effectively extract global information to assist in more accurate color restoration, while coordinate awareness requires only a local receptive field to achieve recovery of coordinate-related distortion types.

## 5 CONCLUSION

This study unveils an innovative end-to-end real-world camera imaging pipeline, surpassing existing methods in both performance and efficiency and pioneering a new benchmark for imaging pipeline design. Through a detailed analysis of the ISP and compression workflows, we introduced a coordinate-independent mapping compression module aimed at optimizing feature compression and tone mapping, alongside a coordinate-aware restore module dedicated to restoring coordinate-specific distortions. The bespoke dataset, crafted for camera imaging pipeline evaluation, sets a robust benchmark, catalyzing future research. Our exhaustive evaluation not only confirms the pipeline's efficacy but also underscores its potential to revolutionize future digital imaging technologies.

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
