# OpenReview forum: "An End-to-End Real-World Camera  Imaging Pipeline"
_acmmm.org/ACMMM/2024/Conference — MM2024 Poster_

### Official Review · Reviewer_P6MZ · 2024-05-16

**Rating:** 3
**Confidence:** 2

**Summary:**

This paper presents an end-to-end camera imaging pipeline called RealCamNet, designed to enhance real-world imaging performance. RealCamNet integrates tasks such as denoising, demosaicing, and tone mapping into a single framework, optimizing them holistically. The authors introduce novel modules, including Coordinate-Aware Distortion Restoration (CADR) and Coordinate-Independent Mapping Compression (CIMC), to address coordinate-dependent distortions and perform image compression. Experimental results show that RealCamNet surpasses baseline methods in accuracy and efficiency.

**Strengths:**

1. The manuscript is well-organized and clearly written.
2. Design an end-to-end imaging network that surpasses the baseline in accuracy and efficiency.

**Limitations:**

1. While integrating tasks like denoising, demosaicing, and tone mapping into a single end-to-end neural network allows for overall optimization, it sacrifices the tunability of individual tasks. This high coupling can make it difficult to fine-tune specific components, potentially leading to suboptimal performance in real-world camera imaging scenarios where precise control, such as in low-light conditions or high dynamic range (HDR) scenes, is needed to maintain image detail and color accuracy.
2. The authors have highlighted the construction of a new dataset as a key contribution. However, after reviewing the paper and supplementary materials, it is not clear whether the dataset will be open-sourced to benefit the camera imaging community. Clarifying the plans for open-sourcing the dataset could enhance the impact of this work.
3. The dataset is constructed using only the Canon M50 camera. This may lead to overfitting to this specific camera's characteristics. In contrast, existing datasets often use multiple cameras to enhance model generalization.
4. The manuscript mixes present and past tenses inconsistently. It is recommended to carefully revise the text to ensure tense consistency throughout the document.

**Suitability:**

2

---

### Official Review · Reviewer_VsKJ · 2024-05-23

**Rating:** 5
**Confidence:** 2

**Summary:**

The manuscript presents RealCamNet, a comprehensive end-to-end deep neural network tailored for enhancing real-world camera imaging systems. This initiative seeks to mitigate prevalent issues in conventional imaging pipelines, such as error propagation and the absence of holistic optimization. Central to RealCamNet are two pivotal modules: the Coordinate-Aware Distortion Restoration (CADR) module, which adeptly corrects coordinate-specific distortions including vignetting and dark shading; and the Coordinate-Independent Mapping Compression (CIMC) module, which efficiently handles tone mapping, denoising, demosaicing, and compression tasks. The integration of these modules enables a unified optimization of the entire imaging process, markedly improving image quality and processing efficiency. Additionally, the authors have commendably developed a novel real-world RAW-RGB dataset to train their model, which has demonstrated enhanced rate-distortion performance and diminished inference latency, as evidenced by their experimental results.

**Strengths:**

1.Novelty: The proposed end-to-end architecture for real-world camera imaging pipelines is novel, addressing several practical challenges in a unified framework.
2.Technical approach: The authors provide a thorough analysis of coordinate-dependent and independent distortions, leading to the design of specialized modules like CADR and CIMC. The joint optimization of the full pipeline is a key strength.
3.Evaluation: The paper includes comprehensive experiments and comparisons with existing methods, demonstrating the superiority of RealCamNet in terms of rate-distortion performance and inference latency.
4.Clarity: The paper is well-written, with clear explanations of the proposed architecture, modules, and methodologies.
5.Applications: The proposed method has practical applications in computational photography, image compression, and various multimedia domains.

**Limitations:**

1.Generalization: The evaluation is primarily focused on the proposed dataset. It would be valuable to assess the model's generalization capabilities on diverse real-world scenarios and datasets.

**Suitability:**

3

---

### Official Review · Reviewer_1Hda · 2024-05-24

**Rating:** 3
**Confidence:** 3

**Summary:**

The paper introduces RealCamNet, an end-to-end camera imaging pipeline designed to enhance real-world camera performance. Unlike traditional multi-stage image signal processing (ISP) methods, RealCamNet integrates all stages into a single framework, enabling joint optimization and addressing coordinate-dependent distortions. The system comprises the Coordinate-Aware Distortion Restoration (CADR) module for distortion correction and the Coordinate-Independent Mapping Compression (CIMC) module for tone mapping and feature compression. A new dataset is also presented to train and evaluate this system.

**Strengths:**

-  Integrating ISP and compression processes into a single pipeline allows for joint optimization, reducing errors and computational redundancy.
- The creation of a comprehensive dataset enhances the training and evaluation of the model, providing a robust benchmark for future research.
- The paper is logically organized with clear sections detailing the methodology.

**Limitations:**

- Although the paper compares RealCamNet with state-of-the-art learning-based methods, it lacks a thorough comparison with advanced non-learning-based ISP methods.
- The use of a single dataset may result in the model being overly tailored to the specific characteristics of that dataset. This can lead to overfitting, where the model performs well on the training and evaluation data but fails to generalize to new, unseen data.
- The evaluation would be significantly strengthened by including comparisons on multiple, publicly available datasets. This would provide a more comprehensive benchmark and allow for a clearer assessment of RealCamNet's performance relative to existing methods.
- In quantitative results comparison, whether other methods are retrained on the proposed dataset?

**Suitability:**

2

---

### Official Review · Reviewer_feXe · 2024-05-27

**Rating:** 3
**Confidence:** 3

**Summary:**

The paper discusses an end-to-end real-world camera imaging pipeline that focuses on the transition from traditional module separation design to fully integrated frameworks like RealCamNet. It explores various image signal processing (ISP) operations such as denoising, demosaicing, tone mapping, and image compression. This method emphasizes the importance of efficient imaging and compression technologies in the digital content era and highlights the need for techniques that reduce file sizes while maintaining image quality.

**Strengths:**

1. The paper introduces RealCamNet, a novel end-to-end framework for camera imaging that categorizes ISP operations into coordinate-independent and coordinate-dependent groups. In this way, the model can address specific distortions, and improve the overall performance of the camera imaging pipeline by the tailoried operations.

2. The paper proposes the Coordinate-Aware Distortion Restoration (CADR) and Coordinate-Independent Mapping Compression (CIMC) modules, which address coordinate-dependent distortions and improve image quality with relatively lower complexity.

**Limitations:**

1. It is essential for the RealCamNet framework to demonstrate adaptability and robustness in handling the inherent variability of real-world imaging conditions. The paper should elaborate on how the CADR module adjusts to different levels of distortion severity and spatial variations encountered in practical settings.

2. While the CADR module is designed to correct coordinate-dependent distortions like vignetting and dark shading, the paper may not provide sufficient technical details on the specific algorithms or methodologies used within the module to address these distortions effectively.

3. In my opinion, the scalability of the RealCamNet framework to handle a wide range of camera models, sensor types, and imaging conditions is not extensively addressed. How to solve it?

**Suitability:**

2

---

### Meta-Review · Area_Chair_1Rwn · 2024-06-27

**Recommendation:** Accept (Poster)
**Confidence:** 4

**Metareview:**

The paper proposes an end-to-end real-world camera imaging pipeline with a focus on the transition from traditional module separation design to fully integrated frameworks.
Several issues have been raised with this paper, such as the quality of experimentation and datasets. These issues should be addressed for the final draft of the paper.